# Trust and transparency in times of crisis: Results from an online survey during the first wave (April 2020) of the COVID-19 epidemic in the UK

**Luisa Enria[1], Naomi Waterlow[2], Nina Trivedy Rogers[3], Hannah Brindle[4], Sham Lal[4], Rosalind M. Eggo[2], Shelley Lees[1], Chrissy h. Roberts[4]***

1 Department of Global Health and Development, Faculty of Public Health & Policy, London School of Hygiene & Tropical Medicine, London, United Kingdom, 2 Department of Infectious Disease Epidemiology, Faculty of Epidemiology & Public Health, London School of Hygiene & Tropical Medicine, London, United Kingdom, 3 UCL Research Department of Epidemiology & Public Health, Faculty of Medical Sciences, University College London, London, United Kingdom, 4 Department of Clinical Research, Faculty of Infectious & Tropical Diseases, London School of Hygiene & Tropical Medicine, London, United Kingdom

☯ These authors contributed equally to this work.
* chrissy.roberts@LSHTM.ac.uk

## Abstract

### Background

The success of a government's COVID-19 control strategy relies on public trust and broad acceptance of response measures. We investigated public perceptions of the UK government's COVID-19 response, focusing on the relationship between trust and perceived transparency, during the first wave (April 2020) of the COVID-19 pandemic in the United Kingdom.

### Methods

Anonymous survey data were collected (2020-04-06 to 2020-04-22) from 9,322 respondents, aged 20+ using an online questionnaire shared primarily through Facebook. We took an embedded-mixed-methods approach to data analysis. Missing data were imputed via multiple imputation. Binomial & multinomial logistic regression were used to detect associations between demographic characteristics and perceptions or opinions of the UK government's response to COVID-19. Structural topic modelling (STM), qualitative thematic coding of sub-sets of responses were then used to perform a thematic analysis of topics that were of interest to key demographic groups.

### Results

Most respondents (95.1%) supported government enforcement of behaviour change. While 52.1% of respondents thought the government was making good decisions, differences were apparent across demographic groups, for example respondents from Scotland had lower odds of responding positively than respondents in London. Higher educational levels

**Data Availability Statement:** The data used in this study are available from LSHTM Data Compass at the following DOI https://doi.org/10.17037/DATA.00001860. The quantitative data (https://doi.org/10.17037/DATA.00001851) are available without restriction, whilst the qualitative data (https://doi.org/10.17037/DATA.00001859) contain sensitive individual level data and will require a data sharing agreement. This can be obtained by requesting access to the data through LSHTM Data Compass (https://datacompass.lshtm.ac.uk/1860/)

**Funding:** The author(s) received no specific funding for this work.

**Competing interests:** The authors have declared that no competing interests exist.

saw decreasing odds of having a positive opinion of the government response and decreasing household income associated with decreasing positive opinion. Of respondents who thought the government was not making good decisions 60% believed the economy was being prioritised over people and their health. Positive views on government decision-making were associated with positive views on government transparency about the COVID-19 response. Qualitative analysis about perceptions of government transparency highlighted five key themes: (1) the justification of opacity due to the condition of crisis, (2) generalised mistrust of politics, (3) concerns about the role of scientific evidence, (4) quality of government communication and (5) questions about political decision-making processes.

## Conclusion

Our study suggests that trust is not homogenous across communities, and that generalised mistrust, concerns about the transparent use and communication of evidence and insights into decision-making processes can affect perceptions of the government's pandemic response. We recommend targeted community engagement, tailored to the experiences of different groups and a new focus on accountability and openness around *how* decisions are made in the response to the UK COVID-19 pandemic.

## Introduction

In response to the pandemic spread of Severe Acute Respiratory Syndrome Coronavirus 2 (SARS-CoV-2) (with cases first reported in Wuhan in China's Hubei province in December 2019) governments across the world introduced a diverse range of control measures, varying in stringency and timing of implementation [1]. Interventions have included a spectrum of responses from the predominantly voluntary guidance (eg: Sweden) to broad-ranging and near complete societal lockdowns in some regions of China.

The relative efficacy of different policy decisions have been, and continue to be, debated amongst scientists, decision-makers and the public. Previous epidemics across the world have shown that a key component for the success of any outbreak response measure is the extent of public acceptance of its legitimacy [2–4]. Trust is crucial, but it is also contextual: citizens' experiences of specific interventions and their perceptions of the institutions delivering them are shaped by social, political and economic structures and historical trajectories [5, 6]. At the same time, trust is not static: it can be built or lost over the course of the response. Research during the 2014–2016 West African Ebola outbreak for example showed how local responders employed varied "technologies of trust", such as openness, accountability and reflexivity to respond to on the ground realities and build confidence in the measures implemented to contain the epidemic [7]. As risk communication and community engagement become increasingly recognised as central to global epidemic response strategies, understanding the dynamics of (mis)trust and the factors that influence the legitimacy of various public health measures is key for developing effective interventions [8–10]. In the current Coronavirus disease 2019 (COVID-19) pandemic, as governments have requested, and in some instances strictly enforced, significant behavioural change and sacrifices in the midst of lockdowns and economic slowdown, building trust and buy-in from citizens has been highlighted as a particular challenge [11].

The UK registered its first case of COVID-19 on the 29[th]January 2020 and in the two months that followed, the government implemented a number of increasingly stringent measures, initially delaying lockdown in favour of light-touch recommendations that the population should adopt social distancing and self-quarantine if experiencing symptoms. By 16[th] March, Prime Minister Boris Johnson advised against 'non-essential travel' and contact with others, whilst adults over the age of 70 and those with specific pre-existing conditions received recommendations to 'shield' for at least 12 weeks. The UK entered a nation-wide lockdown on 23[rd] March. Two days later the Coronavirus Act 2020 was passed, giving the government powers that prohibited gatherings and specified police powers to detain and fine people contravening the rules of lockdown. Our survey therefore captured the first period following the implementation of stringent measures (April 2020) and the 'acute' phase as numbers of infections and deaths rose steadily.

Whilst the importance of trust in an effective outbreak response is widely recognised, the determinants of (mis)trust in epidemic response measures are less well understood. In particular, qualitative research in recent epidemics has shown that we need to understand the dynamics of trust as they vary by socio-political context and the specific outbreak [6, 7, 10, 12].

In this paper we explore a particular aspect of the dynamics of public trust in the UK government's response to COVID-19, namely the relationship of perspectives on the transparency of information being made available to the public and participants' evaluations of the government's pandemic response. We expand existing qualitative work in this field by using an embedded mixed methods approach to data analysis that combines statistical analysis, structural topic modelling (STM) and qualitative thematic coding. The paper explores how perceptions of UK government transparency (or lack thereof) influence broader narratives of trust in institutional responses to the COVID-19 pandemic.

## Methods

### Research design

The project was designed by a multidisciplinary team, including anthropologists [13]. This meant embedding qualitative research questions and analysis in a quantitative survey. This multidisciplinary and mixed methods approach allowed us to combine an understanding of general trends in participants' attitudes of trust and perceptions of transparency with qualitative questions about process, allowing participants to expand on their reasons. Quantitative and qualitative approaches were not used independently but rather designed to complement and build on each other. For example, decisions about coding of qualitative responses, as discussed further below, was directed by a first round of statistical analysis and machine learning supported Structural Text Modelling (STM). This enabled us to supplement quantitative data with qualitative explanations whilst also triangulating between datasets.

### Online survey

Anonymous survey data from UK residents were collected online between 2020-04-06 and 2020-04-22 using an ODK XLSForm (https://getodk.github.io/xforms-spec/) deployed on Enketo smart paper (https://enketo.org/) via ODK Aggregate v.2.0.3 (https://github.com/getodk/aggregate). Form level encryption and end-to-end encryption of data transfer were implemented on all submissions. The survey is included in the supporting information as both PDF (S1 File) and XLSForm (S2 File) formats.

The survey included 49 questions which covered a broad range of topics including (1) Demographics, (2) Health and health behaviours, (3) Adherence to COVID-19 control measures, (4) Information sources used to learn about COVID-19, (5) Trust in various

information sources, government and government decision-making, (6) Rumours and misinformation, (7) Contact & communication during COVID-19 and (8) Fear and isolation.

The survey was distributed using Facebook's premium "Boost Post" feature. A "boosted" post functions as an advert which can be targeted at specific demographics. We boosted details of the survey and its URL to a target audience of 113,280 Facebook users aged 13–65+ years and living in England, Wales, Scotland and Northern Ireland. The survey was further distributed using a 'daisy-chaining' approach in which respondents were asked to share and encourage onward sharing of the survey's URL among friends & colleagues. A number of faith institutions, schools and special interest groups were also contacted directly for assistance in dissemination of the URL.

## Trust and transparency

In this paper we focus on five survey questions that, taken together, allowed us to explore the relationship between trust in the UK government's COVID-19 response and perceptions of transparency.

Of these questions: four broadly assessed self-reported trust quantitatively, including assessments of the response and perceptions about prioritisation and acceptability of enforcement of pandemic measures. To gain insights into self-reported levels of trust in the UK government's epidemic response, participants were asked the question Q1: *"Do you think the government is making good decisions about how to control COVID-19?"* (options "Yes" or "No"). To identify how they viewed the government's prioritisation at the start of the pandemic, participants were asked *Q2: "Do you think that the government cares more about people and their health or the economy?"* (options "Don't know", "They care more about people and their health", "They care more about the economy" and "About the same"). Respondents were asked Q3: *"Do you think that it is acceptable for governments to force some people to change their behaviours in order to control COVID-19?"* (options "Yes" or "No").

In order to explore the interplay between trust in the response and the perceptions of transparency we asked Q4: *"Do you think the government tells you the whole truth about coronavirus and COVID-19?"* (options "Always", "Mostly", "Sometimes", "Almost never", "Never" and "I don't know"). Any participant who did not reply "Always" to the latter question was then invited to answer Q5: *"Briefly describe what it is that you think the government is not being fully truthful about"* in an open-ended text response.

## Ethics, confidentiality & participant wellbeing

The study was approved by the London School of Hygiene & Tropical Medicine observational research ethics committee (Ref: 21846). During the survey, participants were asked to provide (voluntarily) the first two letters of their UK postcode, thus allowing the study team to localise respondents to broad geographical "postcode areas". These areas cover on average several hundred thousand individuals. All data were fully anonymous and the study team had no means by which they could identify individual respondents. All participants provided informed consent to participate in the study by ticking a box on the survey web-form after first having read a short passage of information about the study. The LSHTM ethics committee approved a minimal information form, but links were also provided to a project website which included a more detailed background information and study protocol. A copy of the informed consent text is included in the supporting information (S1 File). All questions in the survey were optional (excepting age and number of people in the household), meaning that participants could skip questions if they chose to.

## Statistical analysis

All analysis was performed in R v4.0 and R scripts required to reproduce the statistical methods are included in the supporting information (S3 File). Using the *mice* package in R, we imputed missing data by performing multiple imputation by chained equations, completing 20 imputed datasets for all relevant fields and pooling results of the 20 separate analyses using Rubin's rules. All reported percentages were calculated from valid data of the non-imputed dataset. We used logistic regression (binomial *glm*) to test for associations between demographic factors (age, education, gender, geographical region & income) and data on participants' opinions of the quality of UK government decision-making. Chi-squared analysis was used to test whether there was any significant association between participants' perceptions on the government's truthfulness and their opinions on the quality of government decision making. To estimate the magnitude of these effects we then re-ran the above regression analysis, including the truthfulness variable as an additional explanatory factor. The 'nnet' R package was then used to apply a multinomial log-linear model via neural networks to the detection of factors which were associated with opinions on government response priority (which had three possible outcome classes). These analyses were corrected for all demographic covariates.

## Topic modelling and analysis of qualitative data

Participants who thought that the government was not being fully truthful about coronavirus and COVID-19 were asked to "Briefly describe what it is that [they thought] the government [was] not being fully truthful about". We applied a Structural Text Modelling (STM) [14] approach to identify key topics in these open-ended text data responses. STM employs machine learning (ML) to explore open ended survey questions in a structured and reproducible way [14, 15] and with a goal to identify topics and perspectives in free-text data. Unlike more conventional topic modelling, STM makes it possible to link topic models to metadata [14, 15] and by doing so to identify groups of responses with similar topic content. This analysis was performed using the '*stm*' package [14] for R. The text data were processed into a corpus and numbers, common punctuation, capitalisation and stop-words (such as "I", "me", "that's" and "because") were removed. Only words which appeared in 15 or more responses to the survey were retained. The number of topics was then determined by looking for a balance between semantic coherence (clear and understandable topics) and exclusivity (lack of crossover between topics). The topic modelling was then run and the resulting topics were labelled manually by assessing both key words used within topics and representative quotes. The number of topics and the topic labels were the main subjective parts of the STM. Expected text proportions (ETP) were defined as the proportion of the total corpus which related to each topic.

Survey submissions with no response to the open text question (mostly from those who felt that the government was fully truthful, a group who were not asked to comment in open-text) were excluded from this analysis.

## Qualitative analysis

The qualitative data analysis focused on responses to the question analysed through STM, namely: Q5: *"Briefly describe what it is that [they thought] the government [was] not being fully truthful about"*. As noted above, this question was included to understand perceptions of transparency and our analysis focused on articulations of (mis)trust within these responses. In order to do so, we chose three topics from the STM analysis that we felt would give qualitative insights into the relationship between trust and transparency and which also closely related to our other three quantitative questions that focused on trust as defined above (Q1, Q2, Q3). As such, we conducted in depth thematic coding on topics for responses to Q5 that elaborated on

perceptions of transparency itself (T1: extent of truth) and perceptions of the government's implementation and prioritisation (T5: implementation and T7: rationale/politics). In order to further tease out the relationship between trust and transparency we focused analysis on responses from the social groups that were found to have been statistically most and least likely to positively evaluate the government's decisions on COVID-19.

Thematic coding was conducted and individual codes, as well as consistency between them, were triangulated with the results of the STM modelling. Our experienced research team conferred regularly to check, refine and agree on the final codes.

## Results

### Quantitative analysis

The analysis was based on data provided by 9,322 respondents aged 20 years and over. A *post-hoc* power calculation for logistic regression was used to determine that the sample size of 9,322 gave us 98% power to detect odds ratios greater than 1.1 at alpha = 0.05 for any explanatory classes of frequency 0.1 and above. No appropriate method for power calculation in multinomial logistic regression was available, but we expect that the large sample size and the limited number (three) of outcome classes in the multinomial analysis were sufficient to adequately power the study to detect small effects in all but the most rare explanatory classes; for instance in Black, Asian and Minority Ethnic (BAME) groups, a limitation which we discuss below.

Respondents of the study were predominantly female (78.5%) and aged between 35 and 69 years (81.6%) (Table 1). A substantial percentage of the participants (61.4%) had a university education. The majority of participants were members of a white ethnic group (95.4%) and there was under-representation of BAME participants (4.1%). There was almost universal agreement (96.5%, n = 8,863) amongst respondents that it would be "acceptable for governments to force some people to change their behaviours in order to control COVID-19". When asked whether they thought the government was making good decisions about how to control COVID-19, 52.7% (n = 4,845) answered positively. Self-reported trust in the government's response was not uniform across different demographic groups (Table 2). Compared to participants living in London, those in Scotland had a lower odds (OR 0.71, 95% CI 0.51–0.91, p = 0.001) of thinking that the government was making good decisions. Meanwhile, participants from the East of England, the South East and the West Midlands all had higher odds than Londoners of thinking that the government was making good decisions (Table 2). Increasing educational levels saw a decreasing odds of having a positive opinion of government decisions (Table 2). Similarly, decreasing household income correlated with decreased positivity in this respect. Males and younger adults had relatively lower odds of having a positive opinion of government decision making than the reference groups (females and age 70+, respectively).

There was diversity in the opinion of different demographic groups with respect to whether the UK government strategy prioritised the economy, people & their health, or a balance of both (Fig 1). People living in Scotland (OR 2.18, 95% CI 1.94–2.42, p < 0.001) and Northern Ireland (OR 1.69, 95% CI 1.18–2.20, p = 0.043) had a higher odds ratio of believing that the economy was the priority than those in other areas. The regions which had higher odds than Londoners of thinking that priorities were focussed on people & their health included the East Midlands (OR 1.32, 95% CI 1.06–1.58, p = 0.046), South East (OR 1.23, 05% CI 1.03–1.43, p = 0.046) and West Midlands (OR 1.28, 95% CI 1.04–1.52, p = 0.049). Groups under the age of 70 had higher odds of citing the economy as the priority (S1 Table). Education also played a role and compared to those whose highest educational achievement was O-Levels or GSCEs,

**Table 1. Demographic characteristics of the study population.**

| Variable | Stats / Values | Freqs (% of Valid) | Missing (% total) |
|---|---|---|---|
| Are government making good decisions about COVID-19? | No | 4352 (47.3%) | 125 (1.34%) |
| | Yes | 4845 (52.7%) | |
| What is the government's response priority? | Don't know | 757 (8.2%) | 124 (1.33%) |
| | Economy | 3139 (34.1%) | |
| | People and their health | 1777 (19.3%) | |
| | About the same | 3525 (38.3%) | |
| Extent to which government tells truth about COVID-19 | Never | 503 (5.5%) | 95 (1.02%) |
| | Almost never | 1335 (14.5%) | |
| | Sometimes | 3094 (33.5%) | |
| | Mostly | 3367 (36.5%) | |
| | Always | 535 (5.8%) | |
| | I don't know | 393 (4.3%) | |
| Government trust free text | Provided text response | 7617 (81.7%) | 0 (0%) |
| | No text response. | 1705 (18.3%) | |
| Acceptable to use force to change people's behaviours | No | 324 (3.5%) | 135 (1.45%) |
| | Yes | 8863 (96.5%) | |
| Age | 20–34 | 618 (6.6%) | 0 (0%) |
| | 35–54 | 3307 (35.5%) | |
| | 55–69 | 4295 (46.1%) | |
| | 70+ | 1102 (11.8%) | |
| Education | Completed Primary School | 64 (0.7%) | 344 (3.69%) |
| | GCSE/O-levels | 873 (9.7%) | |
| | A level/Higher | 591 (6.6%) | |
| | Further education | 1945 (21.7%) | |
| | University (first) degree | 2556 (28.5%) | |
| | Post-graduate degree | 2949 (32.9%) | |
| Gender | Female | 7244 (78.5%) | 89 (0.95%) |
| | Male | 1938 (21.0%) | |
| | All other genders | 51 (0.5%) | |
| Income | Less than £15,000 | 1046 (13.1%) | 1319 (14.15%) |
| | £15,000 - £24,999 | 1510 (18.9%) | |
| | £25,000 - £39,999 | 1830 (22.9%) | |
| | £40,000 - £59,999 | 1673 (20.9%) | |
| | £60,000 - £99,999 | 1328 (16.6%) | |
| | More than £100,000 | 616 (7.7%) | |
| Region | East Midlands | 638 (7.1%) | 401 (4.3%) |
| | East of England | 961 (10.8%) | |
| | North East | 606 (6.8%) | |
| | North West | 919 (10.3%) | |
| | Northern Ireland | 87 (1.0%) | |
| | London | 1331 (14.9%) | |
| | Scotland | 591 (6.6%) | |
| | South East | 1484 (16.6%) | |
| | South West | 1052 (11.8%) | |
| | Wales | 491 (5.5%) | |
| | West Midlands | 761 (8.5%) | |

*(Continued)*

**Table 1.** (Continued)

| Variable | Stats / Values | Freqs | Missing |
|---|---|---|---|
| | | (% of Valid) | (% total) |
| Ethnicity | Arabic | 8 (0.1%) | 55 (0.59%) |
| | Asian | 105 (1.1%) | |
| | Black | 20 (0.2%) | |
| | Mixed (Other) | 72 (0.8%) | |
| | Mixed (White/Asian) | 44 (0.5%) | |
| | Mixed (White/Black) | 29 (0.3%) | |
| | Prefer not to say | 78 (0.8%) | |
| | White | 8840 (95.4%) | |
| | Another Ethnic Group | 71 (0.8%) | |

**Table 2. Relative odds of respondents having a positive opinion of UK government decision-making during the COVID-19 lockdown, by demographic group.**

| Variable | Group | OR | p |
|---|---|---|---|
| Region | East Midlands | 1.19 (0.99–1.39) | 0.085 |
| | East of England | 1.26 (1.09–1.43) | 0.008 |
| | London | Ref | - |
| | North East | 1.15 (0.95–1.35) | 0.157 |
| | North West | 0.98 (0.80–1.16) | 0.793 |
| | Northern Ireland | 0.85 (0.41–1.29) | 0.488 |
| | Scotland | 0.71 (0.51–0.91) | 0.001 |
| | South East | 1.26 (1.11–1.41) | 0.003 |
| | South West | 1.08 (0.91–1.25) | 0.369 |
| | Wales | 0.89 (0.68–1.10) | 0.262 |
| | West Midlands | 1.36 (1.17–1.55) | 0.001 |
| Age | 20–34 | 0.77 (0.56–0.98) | 0.012 |
| | 35–54 | 0.65 (0.50–0.80) | <0.001 |
| | 55–69 | 0.76 (0.62–0.90) | <0.001 |
| | 70+ | Ref | - |
| Education | Completed Primary School | 0.49 (-0.04–1.02) | 0.008 |
| | GSCE/O-Levels (ref) | Ref | - |
| | A level/Higher | 0.62 (0.40–0.84) | <0.001 |
| | Further education | 0.63 (0.45–0.81) | <0.001 |
| | University (first) degree | 0.41 (0.24–0.58) | <0.001 |
| | Post-graduate degree | 0.32 (0.15–0.49) | <0.001 |
| Gender | Female (ref) | Ref | - |
| | Male | 0.77 (0.67–0.87) | <0.001 |
| | All other genders | 0.73 (0.15–1.31) | 0.284 |
| Income | Less than £15,000 | 0.51 (0.29–0.73) | <0.001 |
| | £15,000 - £24,999 | 0.56 (0.36–0.76) | <0.001 |
| | £25000 - £39,999 | 0.60 (0.41–0.79) | <0.001 |
| | £40,000 - £59,999 | 0.72 (0.53–0.91) | 0.001 |
| | £60,000 - £99,999 | 0.85 (0.65–1.05) | 0.097 |
| | £100,000+ (ref) | Ref | - |

Model is adjusted for all covariates.

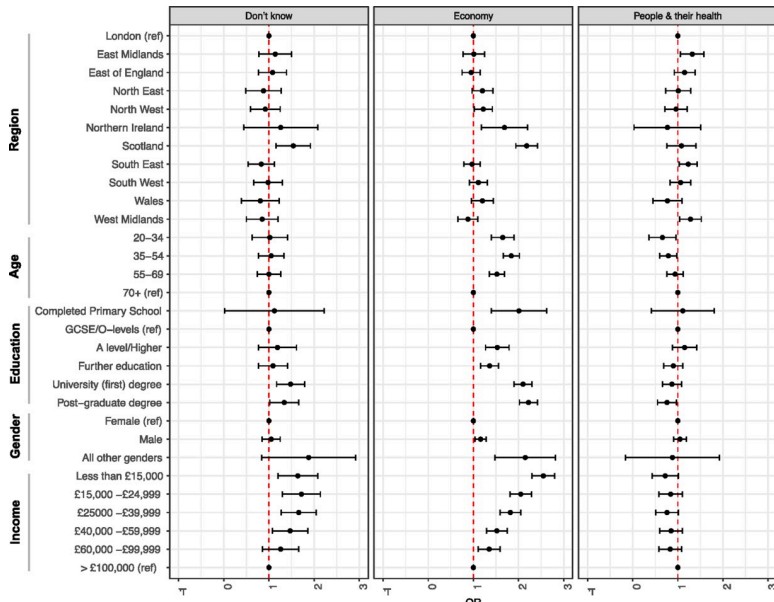

**Fig 1. Participant opinions on UK government prioritisation of COVID-19 response to economy or people and their health.** The statistical model was adjusted for all covariates. Odds ratios compared to those who thought that the priority was a balance of both.

the participants who had A-levels (OR 1.53, 95% CI 1.27–1.79, p = 0.002) or further educational qualifications (OR 1.36, 95% CI 1.16–1.56, p = 0.003) had similar tendency to believe that the focus of response was on the economy rather than on a balanced prioritisation. This effect was stronger still in the group with either a first (OR 2.1, 95% CI 1.9–2.3, p < 0.001) or higher degree (OR 2.22, 95% CI 2.02–2.42), or indeed among the small number of participants who left school after primary education (S1 Table). There was a linear correlation between increasing household income and odds of citing the economy as the government priority (Fig 1).

There was a strong relationship (Fig 2) between responses to the questions about government priority and quality of decision making ($X$-squared = 2999.4, df = 3, p-value < 2.2e-16). Around 60% of participants who thought that the government was not making good decisions also thought that the economy was the priority. 4.7% of this group thought that people and their health were being prioritised, while 25% thought that it was "about the same". In the group who thought more positively of government decisions, 10.17% also thought that the economy was the priority area. 32% thought that the focus was on people and their health compared to 49% who thought that the response took a balanced approach to the two areas.

Around one third (36.5%) of respondents answered that they believed that the government "mostly" told the truth and compared to this group, those who answered 'always' to this question (5.8%) were more likely (OR 2.84, 95% CI 2.47–3.21, p < 0.001) to believe that the government were making good decisions about COVID-19 control. Conversely, those who thought that the government 'never' (0.5% OR 0.03, 95% CI -0.25–0.31, p < 0.001), 'almost never' (14.4% OR 0.03, 95% CI -1.17–0.23, p < 0.001) or 'sometimes' (33.3% OR 0.12, 95% CI 0.00–0.24, p < 0.001) told the truth were all less likely to think that the government was making the good decisions.

## Structural text modelling

STM analysis of the open-text responses resulted in a corpus of 7,589 documents and 786 terms. The model was run with 7 topics until convergence was reached. Seven topics was an

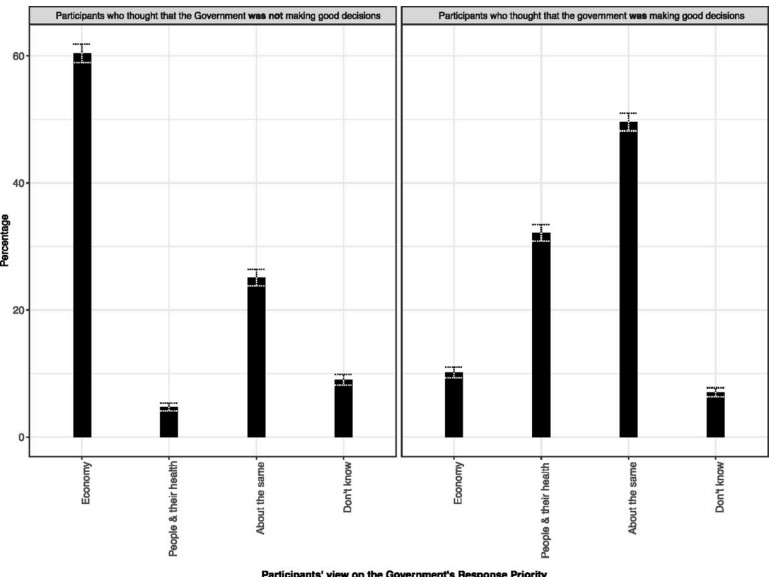

**Fig 2. Perspectives on government prioritisation of health and the economy, stratified by response to the question "Do you think the government is making good decisions about how to control COVID-19?" (Yes/No).** Missing answers were excluded (125 for 'right decision' question, 124 for 'priority' question). Whiskers show 95% confidence intervals.

adequate number based on review of the survey responses and the consistency within topic outcomes. Through analysis of example quotes (S2 Table) and keywords used, topics were manually labelled as (T1) Extent of Truth *[ETP = 0.214]*, (T2) Equipment *[ETP = 0.179]*, (T3) Settings *[ETP = 0.181]*, (T4) Long-term *[ETP = 0.124]*, (T5) Implementation *[ETP = 0.105]*, (T6) Numbers *[ETP = 0.095]* and (T7) Rationale/Politics *[ETP = 0.100]*.

## Qualitative analysis

As discussed above, we then used the results of the STM analysis to focus our in-depth qualitative analysis on a subset of responses that (a) mapped to topics T1: Extent of Truth, T5: Implementation, or T7: Rationale/Politics and (b) came from social groups that were found to have been statistically most and least likely to positively evaluate the government's decisions on COVID-19. These groups were (1) respondents from the UK's devolved nations [Scotland, Wales & Northern Ireland, which have separate legislatures and executives and a range of legal powers that are autonomous of the UK central government at Westminster], (2) respondents who were resident in England with lower and higher levels of education and (3) respondents resident in England and with incomes either under £15,000 or over £100,000.

Responses were then free coded in our qualitative analysis to identify sub-themes that emerged directly from the findings to identify particular narratives, explore qualitative differences between the groups and build a more complex picture of the dynamics of trust within these groups.

Our manual qualitative coding highlighted five major sub-themes that linked together our three chosen topics from the STM analysis for Q5 to produce a coherent qualitative narrative about the relationship between trust and transparency. These sub-themes were similar across groups, though there were differences in the way responses were articulated and in the prominence of particular narratives for different groups.

**Justifying a lack of transparency.** A recurrent theme across all groups included explanations of why the government could not or should not divulge all information about the pandemic. There were four main types of justification in this sub-theme. Firstly, and most commonly, respondents argued that the government had to balance transparency with an avoidance of *"panic"*, *"hysteria"* or *"civil unrest"*. In these kinds of responses, participants emphasised that they did not feel that *"untruthful"* was a correct characterisation, pointing rather to a necessary withholding of information because *"we need to keep a steady hand to come through this one [to] the other side"*. In the University-educated group, some respondents argued that whilst they recognised a need for the government to control the narrative this may have also *"been detrimental to early efforts of containment."*

A second variation of the theme noted that withholding information was necessary to keep a very simple message and to ensure effective behaviour change in the population:

> *"It's not necessarily untruthfulness, I think the government needs to withhold some information to make rules more general. I don't think everyone can be trusted with having enough common sense to curtail their activities and keep social distance for example. So, generalised rules and possibly over-restrictive guidelines are necessary to maintain an average level of obedience."*

This was also expressed in a third variation, namely that government could not share all information because people would not be able to understand it:

> *"I think the information they give is what they think we should know. I think they have to cater for the common denominator. I think they must have sensitive information which the masses don't need to know"*.

Finally, respondents argued that the government could not tell *"the whole truth"* because they likely do not have all the information. Given that *"scientifically, no one really knows what the 'truth' is yet"*, it would be necessary to produce messaging that will ensure citizens abide by the rules:

> *"This is a new disease, so no-one knows the 'whole truth' about it, including the government [. . .] They tell us what they think will lead us to follow their instructions. Truth is not arrived at by democratic vote."*

**Generalised mistrust.** A second set of responses focused on perceptions of a lack of transparency based on overall negative assessments of the government and politics more generally. This finding is important not only in itself, but also because it alerts us to the possibility that attitudes of (mis)trust towards the government's COVID-19 response could be influenced not only by an assessment of pandemic management, but also by broader perceptions and past assessments of the government and public institutions.

Generalised mistrust was articulated for example through broad statements that the government was being untruthful about *"everything"* or: *"I'm not sure [what they are untruthful about] but they have not been truthful about much in the past so it's difficult to believe everything they say now"*. For some respondents this reflected general perceptions that politicians are untruthful, self-interested and intent on prioritising economic interests.

Particularly amongst low-income respondents, this was linked to assessments of the government's track record, with frequent mentions of austerity and underfunding of the NHS.

Mistrust of government was especially common amongst residents of the UK's devolved nations, who expressed dissatisfaction with *"Westminster"* (the central UK government) as a reason for why they felt the government was not telling the whole truth on COVID-19. This was particularly pronounced for Scottish respondents, who contrasted the response of central government with that of the devolved Scottish government in their assessments of transparency and competence:

*"Westminster are not telling the truth, they clearly do not have a clue and they all need to be [held responsible] when this is under control. Scottish government [is] more transparent, faster to react and more all round supportive."*

Welsh respondents linked this more explicitly with central government's track record in their region:

*"I don't trust this government to fully tell the truth. In fact, given their track record over the last ten years, they lie, underfund vital services and appear not to care about the general population. They care about making money and their rich buddies".*

Few respondents 87 (1.0%) were from Northern Ireland, but amongst those there was a disproportionate mention of *"Brexit"* as a factor, suggesting for example that the government's contemporaneous focus on negotiations to leave the European Union may have distracted them when it came to planning a pandemic response.

**Role of evidence.** Although, as noted above, some respondents accepted that existing knowledge about COVID-19 in the initial months of the pandemic was limited, there were significant concerns about what kind of evidence was being used to make decisions and how this evidence was conveyed. Respondents in all groups expressed concern about the balance of science against political or economic considerations. For example:

*"It is not always clear what scientific advice is being given to the government and where this is adjusted by political priorities, in addition the lines between scientific advice, government guidance, opinions of individuals and actual regulations/ legislation are very blurred and not well understood by a lot of the general population."*

Concerns about the role of scientific expertise were particularly prominent amongst respondents with higher education (university degree and above). Comparative word searches between responses from residents in England with higher and lower education backgrounds for example showed that terms like 'science', 'expertise' and 'data' were more frequently cited amongst higher education respondents than lower education respondents (67, 23 and 87 versus 0, 3, 3 respectively).

For example, these concerns were articulated through suggestions that the government *"don't listen to experts"*. University-educated respondents in England gave more specific comments about the kinds of expertise that was either not explained or followed, including:

*"Interpretation of the modelling. Statistics do not always tell the truth"*

In these discussions of evidence, questions about *"herd immunity"* were prominent. Although the government repeatedly denied that it was following a strategy that would see the virus spreading through the population unfettered so as to increase immunity, respondents in the survey who mentioned the controversy believed that this was unofficially the *"overriding policy aim"*. As one respondent put it:

*"I don't believe the government is being transparent about their strategy. I believe they continue to follow their herd immunity strategy as they consider public loss of life acceptable."*

For some respondents this had been the main reason for a loss of confidence in the government: "*After the herd immunity thing, I can't trust them.*"

## Communication

Related to the role of evidence, respondents also expressed their concern with government communication of key information about the pandemic, guidance and strategy. A main strand of discussion was around a perceived lack of coherence and clarity in messaging. This was especially mentioned in relation to seeming contradictions and frequent changes in policy:

*". . .there is sometimes one piece of information one day which is contradicted the next but I think this is mostly scientific and medical experts who are advising the government and who tend to sometimes not agree with each other."*

"*Mixed messages*" and perceptions that risk communication involved "*spin*" or efforts to "*manage*" or "*massage*" the evidence were cited as sources of confusion and mistrust. These narratives envisioned information being "spun" to present the government in a positive light, to obscure mistakes or lack of knowledge about COVID-19. These comments centred especially on the press briefings, which respondents felt delivered "*the agreed message*". The sense that the pandemic response was "*run on slogans*" with "*no detailed information*" meant that the reasoning behind policies and policy changes were not clear.

In contrast, some respondents argued that communication might be forgiven for vagueness and inconsistency, as long as government officials were more open and accountable, for instance by

*"Admitting their mistakes and apologising. We do not expect them to have all the answers and understand if errors have been made but they need to be admitted"*

This was seen to be essential to build confidence: "*Ministers would be well advised to get some help from PR firms who have dealt with crises as to how to really start to build trust.*"

**Decision-making and implementation.** A final group of themed responses centred on a wish for more transparency not only on key statistics, but also on *how* decisions are made. This was particularly pronounced in responses that argued that mistakes in implementation had been made. These mistakes were put down to the unclear role of evidence in defining strategies, the balance of priorities and especially "*political decisions*" and a focus on the economy.

That perception that the government's "*priorities are largely involved in keeping the economy alive and may not involve keeping the number of deaths at a minimum*" was a concern for many respondents. A smaller group, primarily among respondents in the high-income bracket, this was cited as a genuine dilemma: "*I think the government has a difficult job of balancing public health with long term fiscal security.*"

The timing of implementation was a common concern in this sub-theme, with a particular focus on when the lockdown was implemented and future plans to lift restrictions. Initial "*inaction*" and delays in locking down were contrasted with the experience from other countries. Keeping the country open too long, some argued might have been based on political calculations:

> "*[Prime Minister] Boris [Johnson] is the man who said that the real hero in Jaws was the guy who tried to keep the beach open. His own popularity and cabinet over-confidence has come at the expense of following best practice from other counties and medical experts.*"

Other responses around the timing and nature of implementation focused on the preparedness of the NHS, levels of planning at the beginning of the pandemic and the availability of testing and PPE.

## Discussion

Our survey results and mixed methods analysis offer insights into respondents' perspectives of the UK government's COVID-19 response during the first wave of the pandemic and its response in April 2020. In summary, we found that amongst our respondents, there was near unanimous support for government enforcement of behaviour change. Just over half of our respondents thought the government was doing a good job, but this varied across demographic categories, with lower odds for respondents in Scotland, those who were younger, those with higher education and lower income levels. Respondents who did not believe the government was doing a good job were also more likely to believe the economy was prioritised over people and their wellbeing. Around 36% of respondents thought the government mostly told the truth. Amongst those who expressed concerns about a lack of transparency, we found a number of common narratives that offer insights into the relationship between trust and transparency, including reflections on whether a lack of transparency is justified in a time of crisis, deep-seated mistrust in government and concerns about evidence, communication and the politicisation of decision-making.

Our first set of findings relate to overall levels of trust in government decision-making and their leadership in enforcing COVID-19 measures. Political trust, as a "basic evaluative orientation toward the government" [16] is widely recognised as key for the effective functioning of democratic institutions. This becomes ever more important in moments of crisis, including health emergencies, where high levels of uncertainty require confidence in the actors and organisations making decisions about emergency response measures. Before the COVID-19 pandemic, political trust was a major topic of debate amongst political scientists and the public alike against the backdrop of a political crisis triggered by the 2016 referendum to leave the European Union. Analyses of the 'Brexit crisis' highlighted that the referendum reflected long-standing social divisions in the UK and low levels of trust in politicians and institutions [17]. This is in line with trends across Europe and the United States where, in the aftermath of the 2008 financial crisis, confidence in political institutions has steadily declined, with populist 'anti-establishment' parties doing increasingly well electorally [18]. In the UK, the 2019 Eurobarometer survey showed that 21% of respondents said they "tend to trust [the] government to do the right thing", 10 points lower than when the question was asked in 2001 [19]. Against this backdrop, our respondents' evaluations of the UK government's decisions over the first months of the COVID-19 pandemic, whereby 52.7% answered positively, would appear higher than expected. This may suggest that our respondents were more willing to back government decisions at the onset of the crisis. This potential "crisis effect" amongst our respondents is further supported by the fact that there was almost universal agreement that it would be "acceptable for governments to force some people to change their behaviours in order to control COVID-19".

Public acceptance of strong-handed government leadership may increase during times of crisis, particularly in the acute phase of an emergency. Research on counter-terrorism legislation after the 9–11 attacks in the U.S. has shown that 'states of emergency' can affect the

perception of legitimacy of measures that curtail civil liberties in a climate of fear and heightened sense of risk [20]. This work has also pointed to the long-term consequences of these "crisis effects" for democratic values. This literature provides useful parallels for understanding the very high support of government enforcement of behaviour change. This response does not necessarily tell us about respondents' perspectives on whether the government should be able to forcibly change their own individual behaviour, but rather whether enforcement is justified in relation to others. Ignatieff [21] has argued (in the context of counter-terrorism) that majority support for restrictive measures relies on the assumption that these are going to be enforced against a minority who pose a threat to the community at large and that majoritarian acceptance of restrictions on civil liberties plays a role in the securitisation of minorities. In previous epidemics, divisive narratives that distinguished "compliant" citizens and those who were "resistant" individualised responsibility and blame, justifying forcible containment measures that had considerable political consequences [22]. Higher willingness to back the government and acceptance of a need for collective behaviour change are undoubtedly crucial for the outbreak response. However, our participants' responses also reinforce these questions around the broader implications on political rights of the acceptance of strong-handed leadership during moments of crisis.

This is not however the full story, as positive evaluation of the government's COVID-19 decisions was not the same across different groups of respondents. In particular, we found there to be geographical differences, with participants from some of the devolved nations (Scotland in particular) being less likely to evaluate government decisions positively. Income was positively correlated with trust (i.e. wealthier participants were more positive), and education inversely correlated (i.e. more educated participants were less positive) (Table 2). These effects might suggest that pre-existing levels of mistrust are important and previous research on institutional trust (prior to COVID-19) would suggest a positive relationship between income and institutional trust [23]. We might also consider the fact that COVID-19 measures such as lockdowns may have a higher impact on low-income respondents, for instance because of their different experiences of lockdown, differing choices and opportunities with regards to working from home and/or avoiding high risk environments. Such factors could all in turn have an impact on trust. Conversely, the negative relationship between education and trust goes counter to data from OECD research on institutional trust [18]. The higher levels of concern with the role of scientific evidence and expertise cited by respondents in England with higher levels of education suggests that at least for some in this sub-set, observations of the management of the COVID-19 response directly affected their perceptions of transparency and their trust in the government's handling of the pandemic. Respondents who believed that the government was prioritising the economy were more likely to negatively evaluate government decisions (Fig 2). This offers insights into the qualitative dynamics of political trust.

We went on to explore the qualitative mechanisms of (mis)trust in the COVID-19 response, with a particular focus on its relationship to perceptions of transparency. Whilst it is well established that trust is important for both democracy and crisis management in general, *how* trust is achieved, maintained or lost during an emergency is less well understood. Political scientists expect transparency to be an important mechanism, with citizens' ability to access information and to hold governments accountable representing a core pillar of "good governance" [24]. This is particularly pertinent for the COVID-19 pandemic given the attention that has been given to the role of information and misinformation, with the World Health Organization (WHO) alerting to the dangers of an unfolding "infodemic" [25] and political scandals in the UK having influenced popular debate on the topic of good governance during lockdown.

Our findings offer some initial insights on the complex role that transparency plays in citizens' perspectives of the government's response to COVID-19. Whilst 52.7% of respondents

said the government was making the right decisions, only 42.3% thought the government tells the truth about COVID-19 most or all of the time. This appears counterintuitive if we consider common assumptions that transparency is a necessary condition for trustworthy governance. Our qualitative analysis of the free text answers suggests that this gap could be partly explained by some respondents' justification that a lack of government transparency during a crisis is legitimate. These responses argued that governments may have to withhold information in order to prevent panic, because people might not fully understand or because the complexity of the full truth would make it difficult for everyone to comply with guidance. This further supports that there may exist a "crisis effect", which conditions some of our participants' assessments of the UK government's response. This also adds to our previous question about the longer-term impact of emergencies on democratic values, including transparency and accountability, though we found that once again the picture was more complex than it might have initially seemed. Text mining and qualitative coding allowed us to develop a more nuanced analysis of the perspectives amongst our respondents, with a focus on groups who were most and least likely to have a positive perception of the government's response. Generalised mistrust in politics was shared across all groups as a reason for questioning the truthfulness of official information on COVID-19. This suggests firstly that the relationship between trust and perceived transparency is not unidirectional, that is that pre-existing trust in institutions as well as observations of how a pandemic is being managed (in particular how response measures are communicated) can also affect perceptions of transparency. More generally, these expressions of generalised mistrust that pre-existing institutional trust affects attitudes towards an emergency response. This is supported by the fact that responses reflecting low trust in central government, or focusing on the government's track record of defunding public services were particularly prominent amongst low-income, Welsh and Scottish respondents. This points towards both structural and historical determinants of confidence in the epidemic response.

Respondents highlighted a range of other factors that influenced their perceptions of the UK government's response to COVID-19. Across both high-trust and low-trust groups, there were consistent concerns about the coherence, transparency and accountability of communications and decision-making, including uncertainty about the role of evidence and experts, as well as fears that the response was being politicised. This not only gives an insight into the reasons for a lack of trust in the response in low-trust groups, but also suggests that for high trust groups, a positive assessment of government decisions and support for enforcement in a time of crisis did not entirely eliminate concerns about transparency. In the context of debates about misinformation and the role of "fake news" circulating in unregulated communications platforms, our study shows that it is also important to consider trust in official information channels.

## Limitations

Our sample was not population representative and respondents were predominantly white, female and with higher educational attainment. This means that for example, higher levels of trust when compared to pre-crisis levels, could reflect higher levels of structural trust in the sample group. In addition we expect some bias in recruitment towards demographic groups who use Facebook. Because of low uptake, our study was unable to elicit responses from Black, Asian and Minority Ethnic (BAME) communities and we could not draw any conclusions on the perceptions of a demographic group that has been shown to be disproportionately affected by the pandemic [26, 27]. In addition, ethnicity matters for understanding structural levels of political trust, as BAME communities are more likely to experience discrimination and institutional racism across a spectrum of interactions with government [28, 29]. Indeed, as the COVID-19 pandemic develops, ethnic minorities have been shown to be disproportionately

targeted by the enforcement of COVID-19 regulations, including higher rates of fines and arrests. In London, black people were twice as likely to be arrested than white people [30]. This reiterates the importance, as noted above, of exploring the political consequences of epidemic control measures in contexts of structural inequality.

## Recommendations

The extent to which a government may be able to foster public trust in their responses to pandemics appears to be closely linked to the coherence and transparency of their communication strategies. Concerns among UK-based respondents centred around the way the government had used scientific evidence and on how important decisions were made. Based on our findings, we recommend that in order to maintain public trust and acceptance, governments should invest in more transparent, honest governance during pandemics and to provide justification for decisions they make, including the information they cannot share.

Further investigation is required to explore other factors that influence trust in the UK government's response to the COVID-19 pandemic, including the role of personal experiences of disease, levels of trust in the health system, economic and social impacts of the crisis and trust in different kinds of interventions. Comparative analysis across countries will also be able to highlight the relevance of different political structures, histories and relations for the effects of this health emergency on trust and political rights. In addition, our study has only looked at the epidemic's first acute phase of April 2020, and it will be important to continue to explore how perceptions of government performance change over the course of the emergency and beyond. This should include efforts to understand the long-term effects of the COVID-19 crisis on institutional confidence. Maintaining trust is ever more important as the UK transitions in and out of subsequent waves of COVID-19 with fast moving changes to policy, lockdown and other restrictive measures.

Our participants' assessments lead us to reflect on our key finding that there are significant differences in levels of trust across geographical, income and educational backgrounds. Whilst structural determinants of (mis)trust may be hard to act upon in the short-term, it will be important to develop measures such as targeted community engagement that tailor messaging and public deliberation to the realities faced by particular social groups. In contrast to centralised and top-down communication, this approach can directly address the diversity of experiences and perspectives that exist across the country.

Across all demographic groups and regardless of levels of trust, we found that some study participants felt that a lack of transparency was justifiable given the exigencies of crisis. For those respondents who were concerned about transparency, the reasons for those concerns were the same across all groups. Coherent communication, explanations about the sources and roles of different forms of evidence and priorities, a willingness to own up to mistakes and to explain what information cannot be shared could all be practical steps to increasing and maintaining trust across different groups. This would also strengthen accountability beyond the extraordinary times of the COVID-19 emergency.

We speculate that initially very high levels of public acceptance of more draconian control measures may relate to a 'crisis effect' that could be significant, but has the potential to be both acute and short-lived. As the pandemic progresses, governments may not be able to depend on such effects and instead may need to rely on deeper levels of public trust in their strategies to enable them to implement more extreme and restrictive control measures.

## Supporting information

**S1 File. A PDF copy of the survey questionnaire.**
(PDF)

**S2 File. ODK XLSForm design file for the survey questionnaire.**
(XLSX)

**S3 File. R script illustrating analysis.**
(R)

**S1 Table. Participant opinions on UK government prioritisation of COVID-19 response to economy or people & their health.**
(PDF)

**S2 Table. STM topics, expected topic proportions and summaries of thematic content.**
(PDF)

## Acknowledgments

The study team would like to thank the study participants for their contributions to the data and to acknowledge the invaluable contribution of Eleanor Martins' & Esther Amon's administrative support.

## Author Contributions

**Conceptualization:** Luisa Enria, Nina Trivedy Rogers, Hannah Brindle, Rosalind M. Eggo, Shelley Lees, Chrissy h. Roberts.

**Data curation:** Sham Lal, Chrissy h. Roberts.

**Formal analysis:** Luisa Enria, Naomi Waterlow, Chrissy h. Roberts.

**Funding acquisition:** Chrissy h. Roberts.

**Investigation:** Luisa Enria, Nina Trivedy Rogers, Hannah Brindle, Sham Lal, Rosalind M. Eggo, Shelley Lees, Chrissy h. Roberts.

**Methodology:** Luisa Enria, Naomi Waterlow, Nina Trivedy Rogers, Hannah Brindle, Sham Lal, Rosalind M. Eggo, Shelley Lees, Chrissy h. Roberts.

**Project administration:** Chrissy h. Roberts.

**Resources:** Shelley Lees, Chrissy h. Roberts.

**Software:** Nina Trivedy Rogers, Sham Lal, Chrissy h. Roberts.

**Supervision:** Rosalind M. Eggo, Shelley Lees, Chrissy h. Roberts.

**Validation:** Chrissy h. Roberts.

**Visualization:** Rosalind M. Eggo, Chrissy h. Roberts.

**Writing – original draft:** Luisa Enria, Naomi Waterlow, Nina Trivedy Rogers, Hannah Brindle, Rosalind M. Eggo, Shelley Lees, Chrissy h. Roberts.

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
