## [Decision Letter · Decision Letter 0]

6 Jan 2021

PONE-D-20-28524

Trust and Transparency in times of Crisis: Results from an Online Survey During the First Wave (April 2020) of the COVID-19 Epidemic in the UK

PLOS ONE

Dear Dr. Roberts

Thank you for submitting your manuscript to PLOS ONE. After careful consideration, we feel that it has merit but does not fully meet PLOS ONE’s publication criteria as it currently stands. Therefore, we invite you to submit a revised version of the manuscript that addresses the points raised during the review process.

Most importantly, we'd like you to address the comments and/or recommendations made by Reviewers (1, 3 and 4) pertaining to the design/methodology and discussion sections of your manuscript. These reviewers have additionally requested points of clarity and/or made recommendations that will enhance the discussion and conclusions. All the Reviewers' reports are included in this letter. 

Furthermore, there are a few minor grammatical/typographical changes that need to be made.

Structural changes recommended by Reviewers 3 & 4 are for your consideration.

We look forward to receiving your revised manuscript.

Kind regards,

Adriano Gianmaria Duse, MD

Academic Editor

PLOS ONE

Journal Requirements:

2. Please provide additional details regarding participant consent. In the ethics statement in the Methods and online submission information, please ensure that you have specified whether consent was informed.

3. Please provide a sample size and power calculation in the Methods, or discuss the reasons for not performing one before study initiation.

4. Please include additional information regarding the survey or questionnaire used in the study and ensure that you have provided sufficient details that others could replicate the analyses. For instance, if you developed a questionnaire as part of this study and it is not under a copyright more restrictive than CC-BY, please include a copy, in both the original language and English, as Supporting Information.

5. To comply with PLOS ONE submission guidelines, in your Methods section, please provide additional information regarding your statistical analyses. For more information on PLOS ONE's expectations for statistical reporting, please see https://journals.plos.org/plosone/s/submission-guidelines.#loc-statistical-reporting.

6. We note that you have indicated that data from this study are available upon request. PLOS only allows data to be available upon request if there are legal or ethical restrictions on sharing data publicly. For more information on unacceptable data access restrictions, please see http://journals.plos.org/plosone/s/data-availability#loc-unacceptable-data-access-restrictions.

Reviewers' comments:

Reviewer's Responses to Questions

**Comments to the Author**

1. Is the manuscript technically sound, and do the data support the conclusions?

Reviewer #1: Yes

Reviewer #2: Yes

Reviewer #3: Yes

Reviewer #4: Yes

2. Has the statistical analysis been performed appropriately and rigorously? 

Reviewer #1: Yes

Reviewer #2: Yes

Reviewer #3: I Don't Know

Reviewer #4: Yes

3. Have the authors made all data underlying the findings in their manuscript fully available?

Reviewer #1: Yes

Reviewer #2: No

Reviewer #3: Yes

Reviewer #4: Yes

4. Is the manuscript presented in an intelligible fashion and written in standard English?

Reviewer #1: Yes

Reviewer #2: Yes

Reviewer #3: Yes

Reviewer #4: Yes

5. Review Comments to the Author

Reviewer #1: The research covers a topical discussion which is particularly relevant during this pandemic. The sample size is adequate but not rerpesentative of a broad spectrum of the population. ( acknowledged in the report) This is particularly important as the sample largely included educated, white, females - whereas covid affected largely the BAME population - perhaps the study would have produced completely different results had this population group been sampled. The choice of a face book sampling method excludes many of the lower income BAME population as they are less likely to have access to internet and social media ( due to costs). Relying on a sampled person to then forward the invitation to participate would then perpetuate this sample choice. Also females ar more likely to respond than males to online social media requests

Reviewer #2: The authors indicated that the quantitative data are available without restriction, but not the qualitative data, because they contain sensitive information. They provided details on how to request access to this data.

This is a well-written, clear and nicely structured paper. It focuses on very topical and important issues related to government control strategies in the Covid-19 response in the UK. The mixed methods study is well-described and executed, and the methods are appropriate. The results are expertly and clearly presented, and the discussion is enlightening, perceptive and valuable. The limitations are frankly and transparently addressed. Important and helpful recommendations are made.

All in all, it is an excellent paper, based on an important study that makes a very valuable novel contribution to the field.

I am quite comfortable recommending that it be published without any amendments.

Reviewer #3: Overall, a very well-written paper on an incredibly important topic in this pandemic year. I have only a few minor comments. I would ask the authors to better describe the design of the study so that the importance and rigour of mixed methods studies is highlighted. I further appreciate the authors' recognition of the limitations of online surveys.

General comments

• It lacks a section on the design of the study. You mention a quantitative survey and mixed method analysis separately, however this really sounds like an embedded mixed methods design (Creswell, Plano Clark et al 2003) to me. Can you explain this better.

• Would have preferred to see the results per theme rather than per method, integrating both quantitative and qualitative results under thematic headings (with a first paragraph describing Respondent Characteristics). However this is just a preference not an actual requirement. Some of the explanations that are now under Results (for example: the second paragraph under Structural Text Modelling) fits better in the Methodology section. It would have been a more enticing read for me if the complementarity of the qualitative and quantitative data could have been highlighted through such integrated thematic sections.

Minor comments results:

• “were members of a white ethnic group” – do you mean “reported to identify with…”? or how was this assessed?

• Explain “devolved nations” for non-UK residents

• Can you explain more about the rationale underlying these decisions? (in terms of choice of topics and choice of respondents): “We used the results of the STM analysis to focus our in-depth qualitative analysis on a subset of responses that (a) mapped to topics T1: Extent of Truth, T5: Implementation, or T7: Rationale/Politics and (b) came from social groups that were found to have been statistically most and least likely to positively evaluate the government’s decisions on COVID-19. These groups were (1) respondents from devolved nations, (2) respondents who were resident in England with lower and higher levels of education and (3) respondents resident in England and with incomes either under £15,000 or over £100,000.”

Minor comments discussion:

• Would be good to read about why you think respondents with higher education and lower income were less likely to trust the government, and how this relates to the pre-existing institutional trust that you talk about later in the discussion. It is hinted that the consistent underfunding of the NHS plays a role in pre-existing institutional mistrust, but not really how this relates to either low income or high education levels.

• Perhaps more ideas could be elicited from a comparative analysis of what highly educated people and lower income respondents said. Overall, a more elaborated description of the differences between the groups selected for the qualitative exploration seems to be lacking. Perhaps there was little difference, but then that qualifies as an interesting result to be explored further in the discussion.

Reviewer #4: A very interesting study! I have identified a few points that need revising.

Abstract:

Mentions trust and transparency. Perhaps clarify that it is perceived transparency.

Methods in abstract need more clarity. Online questionnaire shared through which channels? What type of statistical analyses were performed? Please add one sentence on how each methodological technique contributed towards addressing the main research question.

Conclusion: A typo in the first sentence "different groups experiences." The conclusion could highlight at least one specific interesting finding. For instance, could the authors recommend that the role of scientific evidence be highlighted? Or that such highlights be more obvious when targeting certain groups in society? What would be the one specific finding from this study that the authors hope that readers would retain? Recommend explicitly stating that finding in the conclusion.

Article:

Introduction: "...qualitative research in recent epidemics has shown that we need to understand the

dynamics of trust as they vary by socio-political context and the specific outbreak." Please add citations to support this claim.

How do the authors address the challenge that pre-existing (mis) trust may have shaped the respondents' evaluation of the government's pandemic response and perspectives on the transparency of information being made available to the public? If they acknowledge it, they need to discuss how this process might impact the interpretations of their quantitative results.

Introduction last paragraph, 1st sentence: Suggest re-ordering, or even better, stating in the form of "relationship of X with Y" instead of "between Y and X," where X is theorized as the cause and Y and the effect.

Need to include arguments for why the authors think that perceptions of transparency would influence trust in institutional responses to the pandemic and why not that deep trust in institutional responses to the pandemic (and in general) might influence perception of transparency.

The last sentence in the introduction section needs to be split into two.

Methods:

Please explicitly clarify the dimensions of trust that were investigated. The methods section description of the questions that were asked in order to assess trust and transparency do not make it clear which question is considered as measuring transparency. Need a clear description of which questions measure trust and which measure transparency.

Need more detail in qualitative coding. Any steps taken to strengthen the reliability of the coding? And to triangulate the findings?

Table 2: Title, typo. "respodents"

6. PLOS authors have the option to publish the peer review history of their article (what does this mean?). If published, this will include your full peer review and any attached files.

Reviewer #1: No

Reviewer #2: No

Reviewer #3: No

Reviewer #4: No

---

## [Author Response · Author response to Decision Letter 0]

25 Jan 2021

Response to reviewers’ comments 

Editor’s comments

and

Response : We have changed the manuscript structure so that it now adheres to these guidelines.

2. Please provide additional details regarding participant consent. In the ethics statement in the Methods and online submission information, please ensure that you have specified whether consent was informed.

Response : Thanks for highlighting this omission. We have now provided a statement on this in the section on consent and have additionally included a citation to the new supporting information file S1. This file is a full text PDF copy of the survey that includes the study information page. Please also note that the exact text of the survey information was approved by the LSHTM research ethics committee as being sufficient for the purposes of informed consent in this study. 

3. Please provide a sample size and power calculation in the Methods, or discuss the reasons for not performing one before study initiation.

Response : We have now added a power calculation for the logistic regression, which demonstrates that we have very good power to detect effects greater than odds ratio = 1.1. We used multinomial regression as part of the work and it is hard to conceptualise an appropriate power calculation for this method. We have therefore added a line to indicate this, but also stating that the very large sample size gives us confidence that we were powered for small effects in all but those variables where some classes were particularly rare (i.e. BAME classes, which are discussed in detail in the section on the study’s limitations).

4. Please include additional information regarding the survey or questionnaire used in the study and ensure that you have provided sufficient details that others could replicate the analyses. For instance, if you developed a questionnaire as part of this study and it is not under a copyright more restrictive than CC-BY, please include a copy, in both the original language and English, as Supporting Information.

Response : We have now included a copy of the survey in two formats. S1 File provides a PDF copy of the survey questions, along with possible answers for qualitative variables. S2 File provides the survey as an XLSForm design file (.xlsx). This format can be used directly in ODK to reproduce the survey and includes full detail of the form structure and logic controls used in the survey.

5. To comply with PLOS ONE submission guidelines, in your Methods section, please provide additional information regarding your statistical analyses. For more information on PLOS ONE's expectations for statistical reporting, please see https://journals.plos.org/plosone/s/submission-guidelines.#loc-statistical-reporting.

Response : We have added substantially more detail to the methods section on statistical analysis and for the purposes of replication by third parties we now provide our R analysis script as supporting information S3 File. 

6. We note that you have indicated that data from this study are available upon request. PLOS only allows data to be available upon request if there are legal or ethical restrictions on sharing data publicly. For more information on unacceptable data access restrictions, please see http://journals.plos.org/plosone/s/data-availability#loc-unacceptable-data-access-restrictions.

Response : After further discussion with our institutional data protection staff, we have been advised that there are no outstanding ethical or consent issues that restrict us from sharing the qualitative data. We have therefore made the entire dataset available through our institutional repository (datacompass.lshtm.ac.uk). The quantitative data can be accessed through https://doi.org/10.17037/DATA.00001851. Qualitative data are available at https://doi.org/10.17037/DATA.00001859. 

Response : This issue has been resolved and all data are now available using the links provided above.

b) If there are no restrictions, please upload the minimal anonymized data set necessary to replicate your study findings as either Supporting Information files or to a stable, public repository and provide us with the relevant URLs, DOIs, or accession numbers. For a list of acceptable repositories, please see http://journals.plos.org/plosone/s/data-availability#loc-recommended-repositories. We will update your Data Availability statement on your behalf to reflect the information you provide.

Response : This issue has been resolved and all data are now available using the links provided above. Thanks for updating the data availability statement on our behalf. 

7. Please include captions for your Supporting Information files at the end of your manuscript, and update any in-text citations to match accordingly. Please see our Supporting Information guidelines for more information: http://protect-eu.mimecast.com/s/B_blCZ01QiMp5ppIXfKdF?domain=journals.plos.org.

Response : We have now added these captions

Reviewers' comments:

Reviewer #1: 

The research covers a topical discussion which is particularly relevant during this pandemic. The sample size is adequate but not representative of a broad spectrum of the population. ( acknowledged in the report) This is particularly important as the sample largely included educated, white, females - whereas covid affected largely the BAME population - perhaps the study would have produced completely different results had this population group been sampled. 

Response : We certainly agree that better representation of BAME communities in our survey would have potentially led to particularly interesting and somewhat different findings, but we do not agree that COVID-19 is a disease that affects mainly the BAME population. Whilst the diagnosis rates and risk of death among individuals from BAME backgrounds are disproportionately higher than that of, for instance, the white female demographic, whites still make up the majority of the population of the UK and in the period of the study, around 83% of all COVID deaths were in white patients. 

The choice of a facebook sampling method excludes many of the lower income BAME population as they are less likely to have access to internet and social media ( due to costs). 

Response : We appreciate this concern and agree that very low income groups may be disadvantaged with respect to internet access. We are however unaware of any specific evidence for substantial issues of inequity for BAME groups in internet access in the UK. Government data from 2019 suggested that more than 90% of all individuals in the UK had recently accessed the internet and that several BAME groups in fact had significantly higher levels of internet use; for instance among the Chinese community 98.6% of individuals had access, whilst among the white group, 90.5% had access. The true situation may be more complicated than the UK.gov data are able to describe, but in the UK context, we expect that engagement of BAME individuals in surveys such as this is a bigger barrier to participation than lack of internet access. Difficulties in the engagement, recruitment and retention of BAME participants to research studies are common throughout health sciences research and this is a problem that is of itself the subject of ongoing research.

Relying on a sampled person to then forward the invitation to participate would then perpetuate this sample choice. Also females are more likely to respond than males to online social media requests

Response : This method of daisy-chained invitational dissemination accounted for only a small minority of responses, with the majority having come from the paid advertisement on Facebook. The advert targeting algorithms used by Facebook are certainly not random (for instance, they iteratively refine their target audience to maximise response, rather than to ensure demographic evenness) and we expect that this is a greater source of skewness in our demographic reach. The response to our survey is certainly biased towards females, but it should be noted that in numerical terms, we did in fact reach nearly 2000 males; providing a very substantial body of data. 

Reviewer #2: 

The authors indicated that the quantitative data are available without restriction, but not the qualitative data, because they contain sensitive information. They provided details on how to request access to this data.

Response : We have addressed this issue in our response to the editor (above)

This is a well-written, clear and nicely structured paper. It focuses on very topical and important issues related to government control strategies in the Covid-19 response in the UK. The mixed methods study is well-described and executed, and the methods are appropriate. The results are expertly and clearly presented, and the discussion is enlightening, perceptive and valuable. The limitations are frankly and transparently addressed. Important and helpful recommendations are made.

All in all, it is an excellent paper, based on an important study that makes a very valuable novel contribution to the field.

I am quite comfortable recommending that it be published without any amendments.

Reviewer #3: 

Overall, a very well-written paper on an incredibly important topic in this pandemic year. I have only a few minor comments. I would ask the authors to better describe the design of the study so that the importance and rigour of mixed methods studies is highlighted. I further appreciate the authors' recognition of the limitations of online surveys.

• It lacks a section on the design of the study. You mention a quantitative survey and mixed method analysis separately, however this really sounds like an embedded mixed methods design (Creswell, Plano Clark et al 2003) to me. Can you explain this better.

Response : Thank you for this very helpful comment. We have now added a section to our methodology on research design, which includes a discussion of our multi-disciplinary team and how an embedded design allowed us to complement quantitative methods with a qualitative dataset on participants’ reasoning and narratives around the relationship between trust and transparency.

• Would have preferred to see the results per theme rather than per method, integrating both quantitative and qualitative results under thematic headings (with a first paragraph describing Respondent Characteristics). However this is just a preference not an actual requirement. Some of the explanations that are now under Results (for example: the second paragraph under Structural Text Modelling) fits better in the Methodology section. It would have been a more enticing read for me if the complementarity of the qualitative and quantitative data could have been highlighted through such integrated thematic sections.

Response : Thanks for this suggestion. When writing the original manuscript, we aimed to write a paper that followed a thematically organised structure such as the one described by the reviewer, but our initial attempts led to a rather messy narrative that somewhat jumped around. We feel that the way we currently present the data shows the work in a more intuitive way, leading from the quantitative statistical analysis, through the hybrid qual-quant STM, to the fully qualitative text coding and interpretation. In places (such as in the second paragraph of results on the STM analysis) we made decisions to include some passages in the results because the methodological step required reference to the topic names or other data that were derived from preceding steps in the analysis. By including these things in the results, we hoped to strengthen the clarity of the work and to prevent excessive need to keep referring back to the methods at key points such as this. 

Minor comments results:

• “were members of a white ethnic group” – do you mean “reported to identify with…”? or how was this assessed?

Response : The data set is entirely self-reported and includes no variables that have been objectively verified. We have however no reason to doubt the veracity of the participants’ responses with respect to self-reported ethnic, cultural or gender identity; or indeed with respect to any other variable. Our feeling is that if we were to suggest that individuals ‘identified’ as members of a particular ethnic or gender grouping, then this could seem to imply that our team of researchers may not recognise the validity, or factuality, of the identities of our respondents. We prefer therefore to assume that misclassification is rare and to expect in a sample as large as this, that the impact of any incorrect or misclassified data will be very minor. 

• Explain “devolved nations” for non-UK residents

Response : Thanks very much for highlighting how we had assumed a degree of knowledge on the UK’s governmental organisation that many readers would not have. We have now added a parenthesis to identify the UK’s devolved nations and to also explain their semi-autonomy from Westminster. 

• Can you explain more about the rationale underlying these decisions? (in terms of choice of topics and choice of respondents): “We used the results of the STM analysis to focus our in-depth qualitative analysis on a subset of responses that (a) mapped to topics T1: Extent of Truth, T5: Implementation, or T7: Rationale/Politics and (b) came from social groups that were found to have been statistically most and least likely to positively evaluate the government’s decisions on COVID-19. These groups were (1) respondents from devolved nations, (2) respondents who were resident in England with lower and higher levels of education and (3) respondents resident in England and with incomes either under £15,000 or over £100,000.”

Response : Thank you for this comment. We have added the following text to our qualitative analysis section to highlight how we coded one particular open-ended question on transparency and then focused on those topics that allowed us to study directly how attitudes towards transparency were articulated through narratives about trust:

“The qualitative data analysis focused on responses to the question analysed through STM, namely: Q4: “Briefly describe what it is that [they thought] the government [was] not being fully truthful about”. As noted above, this question was included to understand perceptions of transparency and our analysis focused on articulations of (mis)trust within these responses. In order to do so, we chose three topics from the STM analysis that we felt would give qualitative insights into the relationship between trust and transparency and which also closely related to our other three quantitative questions that focused on trust as defined above (Q1, Q2, Q3 and Q5). As such, we conducted in depth thematic coding on topics for responses to Q4 that elaborated on perceptions of transparency itself (T1: extent of truth) and perceptions of the government’s implementation and prioritisation (T5 implementation and T7 rationale/politics).”

“In order to further tease out the relationship between trust and transparency we focused analysis on responses from the social groups that were found to have been statistically most and least likely to positively evaluate the government’s decisions on COVID-19.” 

“Following the statistical analysis, these were identified as residents in devolved nations of Wales, Scotland and Northern Ireland (low trust) and high income/ low education (higher trust) and low income/ high education (lower trust), so we performed our qualitative coding on these 7 sub-sets of open-ended responses.”

Minor comments discussion:

• Would be good to read about why you think respondents with higher education and lower income were less likely to trust the government, and how this relates to the pre-existing institutional trust that you talk about later in the discussion. It is hinted that the consistent underfunding of the NHS plays a role in pre-existing institutional mistrust, but not really how this relates to either low income or high education levels.

Response This is an excellent point that encouraged us to extend our analysis. We have made two key changes. In the section on the role of evidence we have emphasised that this was particularly a concern for English respondents with higher levels of education (qualitative responses by Scottish and Welsh participants were analysed all together and were not differentiated by education and income level). In the discussion then we proposed some interpretations. Whilst lower levels of incomes were already associated with lower levels of trust prior to COVID-19, education is normally expected to have a positive relation to institutional trust. This, combined with a greater concern for the transparent use of scientific evidence, suggests that for people with higher levels of education their observations of the management of the covid-19 response affect their levels of trust regardless of pre-existing institutional trust. 

• Perhaps more ideas could be elicited from a comparative analysis of what highly educated people and lower income respondents said. Overall, a more elaborated description of the differences between the groups selected for the qualitative exploration seems to be lacking. Perhaps there was little difference, but then that qualifies as an interesting result to be explored further in the discussion.

Response: As above, we have now included this particular distinction. In the introduction to the qualitative analysis section we also highlight that though overall responses were quite similar, under each theme we highlight where responses were more prominent for one or more sub-groups of respondents. 

Reviewer #4: 

A very interesting study! I have identified a few points that need revising.

Abstract:

Mentions trust and transparency. Perhaps clarify that it is perceived transparency.

Response - Thanks for making this important distinction. We have added text to the abstract to make it clear that we studied perceived and not objectively measured transparency. 

Methods in abstract need more clarity. Online questionnaire shared through which channels? What type of statistical analyses were performed? Please add one sentence on how each methodological technique contributed towards addressing the main research question.

Response : We have now added some more detail to the abstract as requested. 

Conclusion: A typo in the first sentence "different groups experiences." 

Response : We could not find this error in our draft, but have revised the sentence to have a clearer grammatical structure and hopefully it is improved by the change. 

The conclusion could highlight at least one specific interesting finding. For instance, could the authors recommend that the role of scientific evidence be highlighted? Or that such highlights be more obvious when targeting certain groups in society? What would be the one specific finding from this study that the authors hope that readers would retain? Recommend explicitly stating that finding in the conclusion.

Response : We have added a summary sentence to the section on recommendations, before extending the more nuanced discussion of findings. 

Article:

Introduction: "...qualitative research in recent epidemics has shown that we need to understand the

dynamics of trust as they vary by socio-political context and the specific outbreak." Please add citations to support this claim.

Response : Thank you, these have now been added

How do the authors address the challenge that pre-existing (mis) trust may have shaped the respondents' evaluation of the government's pandemic response and perspectives on the transparency of information being made available to the public? If they acknowledge it, they need to discuss how this process might impact the interpretations of their quantitative results.

Response : In our qualitative analysis we discuss ‘generalised mistrust’ – identifying how some of the responses directly refer to this mistrust. We have added a further sentence to that section to highlight that this kind of pre-existing disposition towards government may also influence other types of responses—i.e. Respondents may be more likely to be suspicious of the government if trust is low. In other words, we are not claiming that trust is simply related to how a response is managed, and our qualitative data shows that pre-existing disposition towards government matters.

Introduction last paragraph, 1st sentence: Suggest re-ordering, or even better, stating in the form of "relationship of X with Y" instead of "between Y and X," where X is theorized as the cause and Y and the effect.

Response : We have made some changes to the paragraph that we hope address this point appropriately. 

Need to include arguments for why the authors think that perceptions of transparency would influence trust in institutional responses to the pandemic and why not that deep trust in institutional responses to the pandemic (and in general) might influence perception of transparency.

Response: Thank you for this very important point. We have qualified this in the discussion to show that qualitative findings support that the relationship is not unidirectional, but rather that trust may also impact perceptions of transparency. This is further supported by the fact that we analysed perceptions of trust across groups with high and low levels of reported trust.

The last sentence in the introduction section needs to be split into two.

Response : This is now done. 

Methods:

Please explicitly clarify the dimensions of trust that were investigated. The methods section description of the questions that were asked in order to assess trust and transparency do not make it clear which question is considered as measuring transparency. Need a clear description of which questions measure trust and which measure transparency.

Response : Thank you for this, we have now clarified this in the section on Trust and Transparency. 

Need more detail in qualitative coding. Any steps taken to strengthen the reliability of the coding? And to triangulate the findings?

Response : This has now been added to the section on qualitative analysis. The STM process uses an iterative expectation maximisation algorithm to classify the topics. It works consistently across multiple runs. Because the coding is almost entirely performed without human interventions, it is also relatively resistant to bias, at least with regards to grouping subsets of responses together as belonging to a specific topic. 

Table 2: Title, typo. "respodents"

Response : Thanks, though we couldn’t find this one in our draft.

---

## [Editor Report · Decision Letter 1]

27 Jan 2021

Trust and Transparency in times of Crisis: Results from an Online Survey During the First Wave (April 2020) of the COVID-19 Epidemic in the UK

PONE-D-20-28524R1

Dear Dr. C Roberts

We’re pleased to inform you that your manuscript has been judged scientifically suitable for publication and will be formally accepted for publication once it meets all outstanding technical requirements.

Kind regards,

Adriano Gianmaria Duse, MD

Academic Editor

PLOS ONE
---

## [Editor Report · Acceptance letter]

1 Feb 2021

PONE-D-20-28524R1 

Trust and transparency in times of crisis: Results from an online survey during the first wave (April 2020) of the COVID-19 epidemic in the UK 

Dear Dr. Roberts:

I'm pleased to inform you that your manuscript has been deemed suitable for publication in PLOS ONE. Congratulations! Your manuscript is now with our production department. 

Kind regards, 

on behalf of

Dr. Adriano Gianmaria Duse 

Academic Editor

PLOS ONE